# Global Structure-Aware Diffusion Process for Low-Light Image Enhancement

Jinhui Hou[1], Zhiyu Zhu[1], Junhui Hou[1],[*] Hui Liu[2], Huanqiang Zeng[3], and Hui Yuan[4]

[1]City University of Hong Kong, [2]Caritas Institute of Higher Education
[3]Huaqiao University, [4]Shandong University

## Abstract

This paper studies a diffusion-based framework to address the low-light image enhancement problem. To harness the capabilities of diffusion models, we delve into this intricate process and advocate for the regularization of its inherent ODE-trajectory. To be specific, inspired by the recent research that low curvature ODE-trajectory results in a stable and effective diffusion process, we formulate a curvature regularization term anchored in the intrinsic non-local structures of image data, i.e., global structure-aware regularization, which gradually facilitates the preservation of complicated details and the augmentation of contrast during the diffusion process. This incorporation mitigates the adverse effects of noise and artifacts resulting from the diffusion process, leading to a more precise and flexible enhancement. To additionally promote learning in challenging regions, we introduce an uncertainty-guided regularization technique, which wisely relaxes constraints on the most extreme regions of the image. Experimental evaluations reveal that the proposed diffusion-based framework, complemented by rank-informed regularization, attains distinguished performance in low-light enhancement. The outcomes indicate substantial advancements in image quality, noise suppression, and contrast amplification in comparison with state-of-the-art methods. We believe this innovative approach will stimulate further exploration and advancement in low-light image processing, with potential implications for other applications of diffusion models. The code is publicly available at https://github.com/jinnh/GSAD.

## 1 Introduction

Low-light image enhancement (LLIE) aims to improve the visibility and contrast of the image captured in poor lighting while preserving the natural-looking details, which contributes to many downstream applications, such as object detection [15, 58] and semantic segmentation [62, 48].

Traditional works employ some techniques, such as histogram equalization [28], Retinex theory [6, 23], and gamma correction [29], to correct the image illumination. In recent years, owing to the powerful representational ability and large collections of data, a considerable number of deep learning-based methods [53, 54, 63, 5, 21, 64, 43, 51, 40, 65] have been presented to significantly improve the performance of low-light enhancement by learning the mapping between low-light and normal-light images. Generally, most of the existing works tend to adopt pixel-wise objective functions to optimize a deterministic relationship. Consequently, such regularization frequently produces suboptimal reconstructions for indeterminate regions and poor local structures, resulting in visibly lower reconstruction quality. Although the adversarial loss might mitigate this issue, these methods necessitate careful training adjustments, which might lead to overfitting on specific features or data distributions and even creating new content or artifacts. Recently, the popular diffusion

---

[*]Corresponding author (Email: jh.hou@cityu.edu.hk). Jinhui Hou and Zhiyu Zhu contributed to this work equally.

37th Conference on Neural Information Processing Systems (NeurIPS 2023).

| **Algorithm 1** Forward process | **Algorithm 2** Reverse process |
|---|---|
| 1: **Repeat** 
 2: $\mathbf{X}_0 \sim q(\mathbf{X}_0)$ 
 3: $t \sim \text{Uniform}(\{1, 2, ..., T\})$ 
 4: $\epsilon \sim \mathcal{N}(0, \mathbf{I})$ 
 5: Take gradient descent step on 
 6: $\quad \nabla_\theta \| \epsilon - \epsilon_\theta(\sqrt{\overline{\alpha}_t}\mathbf{X}_0 + \sqrt{1 - \overline{\alpha}_t}\epsilon, t)\|^2$ 
 7: **Until** converged | 1: $\mathbf{X}_T \sim \mathcal{N}(0, \mathbf{I})$ 
 2: **For** $t = T, ..., 1$ **do** 
 3: $\quad \mathbf{z} \sim \mathcal{N}(0, \mathbf{I})$ if $t > 1$, else $\mathbf{z} = 0$ 
 4: $\quad \mathbf{X}_{t-1} = \frac{1}{\sqrt{\alpha_t}}(\mathbf{X}_t - \frac{1-\alpha_t}{\sqrt{1-\overline{\alpha}_t}}\epsilon_\theta(\mathbf{X}_t, t)) + \sigma_t \mathbf{z}$ 
 5: **End for** 
 6: **Return** $\mathbf{X}_0$ |

denoising probabilistic model (DDPM) [7] has garnered notable interest in low-level vision domains [9, 32], credited with its outstanding capacity in modeling the distribution of image pixels. However, a straightforward implementation of the diffusion model for low-light image enhancement is insufficient to address this issue.

In this paper, we present a novel diffusion-based method to boost the performance of low-light enhancement from the perspective of regularizing the ODE-trajectory. Drawing inspiration from the evidence that suggests the potential of a low-curvature trajectory in yielding superior reconstruction performance [12], we endeavor to modulate the ODE-trajectory curvature. Specifically, we introduce a global structure-aware regularization scheme into the diffusion-based framework by gradually exploiting the intrinsic structure of image data. This innovative constraint promotes structural consistency and content coherence across similar regions, contributing to a more natural and aesthetically pleasing image enhancement while preserving the image's fine details and textures. In addition, we devise an uncertainty-guided regularization by integrating an uncertainty map into the diffusion process, facilitating adaptive modulation of the regularization strength. Experimental results on a comprehensive set of benchmark datasets consistently demonstrate the superior performance of our proposed method compared to state-of-the-art ones. A thorough ablation study emphasizes the significance of both the global structure-aware regularization and uncertainty-guided constraint components in our approach, revealing their synergistic effects on the overall quality of enhancement.

## 2   Related work

**Low-light image enhancement.** The early works tackle low-light image enhancement by applying some traditional techniques like histogram equalization [28, 36], gamma correction [29], and Retinex theory [6, 23]. Some researchers also seek to improve the visibility using additional sensors[67, 66, 42, 41, 37, 17]. In recent years, with the advancement of low-light data collection [11, 6, 45, 14, 54] a significant number of deep learning-based methods [2, 60, 8, 50, 53, 54, 63, 5, 21, 64, 43, 51, 40] have been proposed, which greatly improved the restoration quality of traditional methods. For example, Retinex-based deep learning methods [45, 60, 59, 47] employed deep learning to decompose low-light images into two smaller subspaces, i.e., illumination and reflectance maps. Wang *et al.* [43] proposed a normalizing flow-based low-light image enhancement approach that models the distributions across normally exposed images. Xu *et al.* [51] incorporated the Signal-to-Noise-Ratio (SNR) prior to achieve spatial-varying low-light image enhancement. Wang *et al.* [40] proposed a transformer-based low-light enhancement method. We refer readers to [13] for a comprehensive review of this field.

**Diffusion-based image restoration.** Recently, diffusion-based generative models [34] have delivered astounding results with the advancements in denoising diffusion probabilistic models (DDPM) [7, 24], making them increasingly influential in low-level vision tasks, such as image super-resolution [9, 32], image inpainting [30, 19], image deraining [26], and image deblurring [46]. Saharia *et al.* [32] employed a denoising diffusion probabilistic model to image super-resolution by conditioning the low-resolution images in the diffusion process. Lugmayr *et al.* [19] proposed a mask-agnostic approach for image inpainting by leveraging the pre-trained unconditional DDPM as the generative prior. Ozan *et al.* [26] developed a patch-based diffusion model for weather removal that utilized a mean estimated noise-guided sampling update strategy across overlapping patches. These models are based on a diffusion process that transforms a clean image into a noisy one during training time, and then reverses that Markov Chain to generate new images in the testing phase. Consider a clean image $\mathbf{X}_0 \in \mathbb{R}^{h \times w}$ and a noise variable $\epsilon \in \mathbb{R}^{h \times w} \sim \mathcal{N}(0, \sigma^2 I)$. The training steps of a diffusion process act as Algorithm 1.

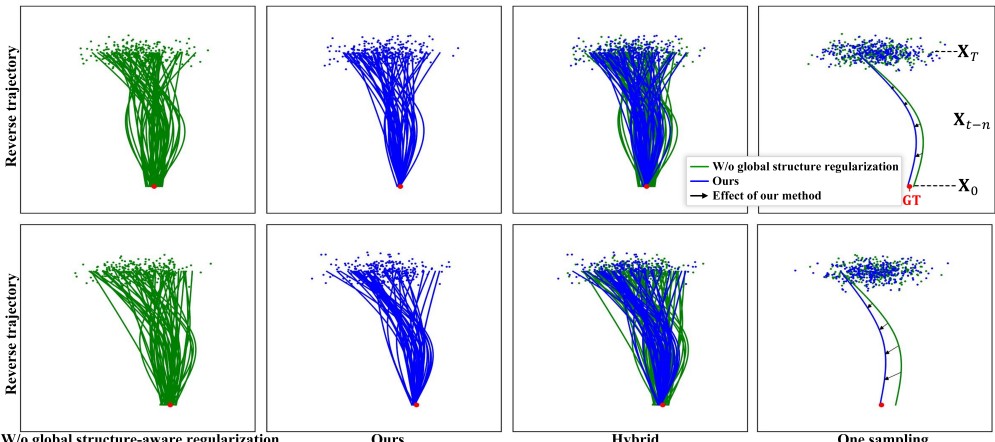

Figure 1: Reverse trajectories of diffusion model with and without our global structure-aware regularization on the testing set of LOLv1 dataset. Our method effectively compactifies the distribution of reverse trajectories of multiple samplings, producing low-curvature reverse trajectories, which makes them stably approach the Ground Truth (GT) values.

Note $\sqrt{\bar{\alpha}_t}\mathbf{X}_0 + \sqrt{1 - \bar{\alpha}_t}\epsilon$ is a closed-form noisy sample at timestamp $t$, and $\epsilon_\theta(\cdot, \cdot)$ denotes a noise estimation network, which perceives the closed-form samples with timestamps and predicts the inherent noise in such samples. By applying this process in reverse, shown as Algorithm 2, from $\mathbf{X}_T \sim \mathcal{N}(0, \mathbf{I})$ to $\mathbf{X}_0$, we obtain a generated image $\mathbf{X}_0$, where $\mathbf{X}_t \in \mathbb{R}^{h \times w}$ is the intermediate reconstructed sample at timestamp $t \in [0, 1, ..., T]$, $\alpha_t$ denotes a scaling scalar, usually parameterized as a linearly decreasing sequence [7, 32], and $\bar{\alpha}_t = \prod\limits_{i=1}^{t} \alpha_i$.

## 3 Proposed Method

### 3.1 Problem Statement and Overview

Low-light image enhancement is a crucial research area in the field of image processing and computer vision, owing to the increasing demand for high-quality images captured in various low-light environments. Mathematically, the degradation of an image captured under low-light conditions can be modeled as

$$\mathbf{Y} = \mathbf{X}_0 \odot \mathbf{S} + \mathbf{N}, \tag{1}$$

where $\mathbf{Y} \in \mathbb{R}^{H \times W \times 3}$ is the observed low-light image of dimensions $H \times W$, $\mathbf{X}_0 \in \mathbb{R}^{H \times W \times 3}$ is the latent high-quality image under normal-light conditions, $\mathbf{S} \in \mathbb{R}^{H \times W \times 3}$ represents the spatially varying illumination map that accounts for the uneven lighting, and $\mathbf{N} \in \mathbb{R}^{H \times W}$ is the noise term introduced due to sensor limitations and other factors. Thus, low-light image enhancement, particularly in reconstructing extreme low-light regions, is of paramount challenge due to the scarcity of available content in such images. The intricacy is further intensified by the content-dependent nature of optimal lighting conditions, necessitating a flexible and adaptive solution. Fortunately, diffusion models, celebrated for their remarkable capacity to synthesize images conditioned on ancillary images or textual cues, emerge as a propitious candidate for tackling the low-light image enhancement conundrum.

In this paper, we advocate capitalizing on the potent generalization capabilities inherent in diffusion models to overcome the obstacles associated with low-light image enhancement. By incorporating low-light images ($\mathbf{Y}$) as conditioning inputs into the diffusion models, we can obtain a naïve implementation of low-light enhancement diffusion model ($\epsilon_\theta(\mathbf{Y}, \mathbf{X}_t, \bar{\alpha}_t)$). Nonetheless, the performance achieved by these preliminary models remains unsatisfactory. In pursuit of procuring high-quality diffusion processes, various strategies have been presented, including alterations to the network structure [31], the incorporation of data augmentation techniques [38], and the introduction of innovative loss terms [22]. Among them, trajectory rectification emerges as a promising strategy, facilitating a seamless and stable diffusion process through the regularization of a low curvature reverse trajectory

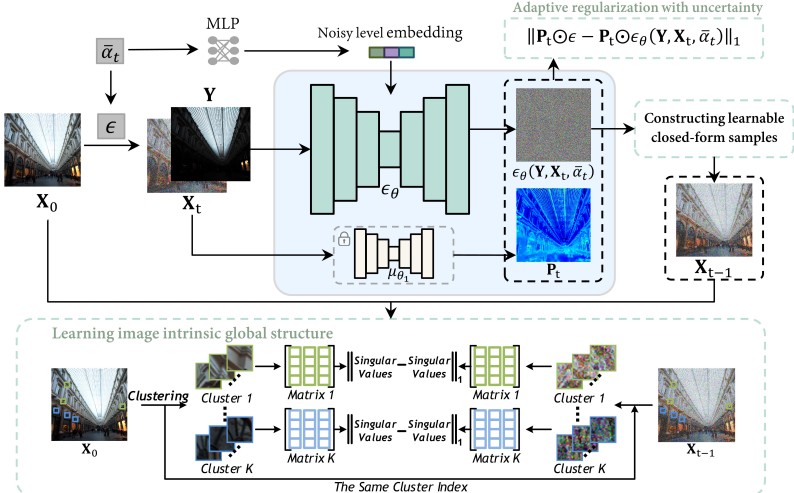

Figure 2: Illustration of the training workflow of the proposed method. We begin by obtaining a closed-form sample $\mathbf{X}_t$ from $\mathbf{X}_0$ and the noise level $\bar{\alpha}_t$, which is subsequently fed into a U-Net $\epsilon(\cdot)$ conditioned with the corresponding low-light image $\mathbf{Y}$ and $\bar{\alpha}_t$ to estimate the employed noise $\epsilon$. To comprehensively regularize reconstructed images, we proceed to construct a learnable closed-form sample $\mathbf{X}_{t-1}$ using the estimated noise $\epsilon_\theta(\mathbf{Y}, \mathbf{X}_t, \bar{\alpha}_t)$ and optimize the model using uncertainty-guided noise estimation and intrinsic image global structure. Meanwhile, we use a pre-trained uncertainty estimation model $\mu_{\theta_1}$ to obtain the uncertainty map $\mathbf{P}_t$.

[12, 18]. Encouraged by current successes, we endeavor to augment this preliminary model from the following two distinct perspectives.

**1) Learning low curvature trajectories with global structure-aware regularization.** To learn a low curvature reverse trajectory, we propose a simple yet effective way via directly minimizing the gap between a learnable sample on the reverse trajectory and the ground-truth sample. This process is further elucidated through pertinent experimental results presented in Fig. 1. Moreover, it is crucial to acknowledge that images intrinsically encompass analogous textures or patterns distributed across different locations [1, 27, 3, 4, 49, 16, 56]. As a result, cultivating a diffusion process that adeptly captures such global structures is indispensable to the success of low-light image reconstruction. By integrating the matrix rank-based regularization within the diffusion-based framework, we astutely exploit the intrinsic structure of image data, thereby facilitating the preservation of nuanced details and the enhancement of contrast.

Despite the effectiveness of the trajectory regularization, precipitating it during the beginning steps of diffusion models could unintentionally subvert sample diversity and quality, attributable to the pronounced fluctuations in large components. Consequently, in pursuit of enhancing the effectiveness of our model, it becomes imperative to architect a *progressively adaptive* regularization schedule.

**2) Adaptive regularization with uncertainty.** Furthermore, most contemporary loss functions treat all pixels equally in the field of low-light enhancement tasks, regardless of whether they exist in conditions of slightly low-light or extremely low-brightness, or whether they are subject to noise-deterioration or are noise-free. Therefore, drawing inspiration from [25], we introduce an uncertainty-guided regularization into the diffusion process to further improve restoration learning in challenging regions.

Those strategic incorporations ensure the precise and resilient recovery of image details in struggling low-light regions while accommodating scene-specific lighting variations. In the following sections, we give technical details of our method.

### 3.2 Exploring Global Structures via Matrix Rank Modeling

The denoising diffusion probabilistic models exhibit a unique learning scheme, which involves learning the distribution of latent noise during training. This characteristic makes it challenging to regularize network parameters $\theta$ for learning global structure-aware representations. To address this

issue, we first construct a learnable closed-form sample of $\mathbf{X}_{t-1}$. Subsequently, we implement two further steps to enhance the content-aware regularization in those images.

**1) Constructing learnable closed-form samples.** The training of diffusion models commences with the procurement of a closed-form $\mathbf{X}_t$ at an arbitrary timestep $t$, as delineated in Line 6 from Algorithm 1. In the ensuing step, a learnable function $\epsilon_\theta(\mathbf{Y}, \mathbf{X}_t, \bar{\alpha}_t)$ is employed to estimate the latent noise $\epsilon$. Given that the network can successfully learn this noise, it becomes feasible to progressively eliminate the noise, as demonstrated by:

$$\mathbf{X}_{t-1} = \frac{1}{\sqrt{\alpha_t}}(\mathbf{X}_t - \frac{1 - \alpha_t}{\sqrt{1 - \bar{\alpha}_t}}\epsilon_\theta(\mathbf{Y}, \mathbf{X}_t, \bar{\alpha}_t)) + \sigma_t \mathbf{Z}. \tag{2}$$

Consequently, throughout the training phase, the regularization is strategically applied to the learnable sample $\mathbf{X}_{t-1}$ in lieu of the closed-form sample $\mathbf{X}_t$. This approach seeks to optimize the performance of the model by focusing on regularization where it can most effectively influence the learning process.

**2) Non-local patch-based matrix rank modeling.** To find global structures of a clean image $\mathbf{X}_0$, we first patchify it into a set of non-overlapping blocks $\{\mathbf{x}_0^{(i)}\}_{i=1}^n$, where $\mathbf{x}_0^{(i)} \in \mathbb{R}^m$ denotes the vectorization of a small block from $\mathbf{X}_0$. To capture the intrinsic structures from the sample $\mathbf{X}_0$, we then apply the commonly-used clustering algorithm to get $k$ clusters of $\{\mathbf{x}_0^{(i)}\}_{i=1}^n$ as $\left\{\{\mathbf{x}_0^{(i)}\}_{i=1}^{n_j}\right\}_{j=1}^k$. It is worth noting that our method benefits from advanced clustering algorithms greatly (See Sec. 4.3 for details). Through stacking each cluster of vectors into a matrix, we could get $\{\mathbf{M}_0^{n_j}\}_{j=1}^k$, where the rank of matrix $\mathbf{M}_0^{n_j} \in \mathbb{R}^{m \times n_j}$ is an effective measurement of the global structures. Consequently, to regularize structure consistency in $\mathbf{X}_{t-1}$ and $\mathbf{X}_0$, we then apply the same aggregation to the learnable sample $\mathbf{X}_{t-1}$, resulting in $\{\mathbf{M}_{t-1}^{n_j}\}_{j=1}^k$, and regularize

$$\mathcal{L}_s(\mathbf{M}_{t-1}^{n_j}, \mathbf{M}_0^{n_j}) = \|\text{diag}(\mathbf{S}_{t-1}^j) - \text{diag}(\mathbf{S}_0^j)\|_1, \mathbf{M}_t^{n_j} = \mathbf{U}_t^j \mathbf{S}_t^j \mathbf{V}_t^j, \tag{3}$$

where $\mathcal{L}_s(\mathbf{M}_{t-1}^{n_j}, \mathbf{M}_0^{n_j})$ indicates the regularization function, $\mathbf{U}_t^j \in \mathbb{R}^{m \times m}$, $\mathbf{S}_t^j \in \mathbb{R}^{m \times n_j}$, and $\mathbf{V}_t^j \in \mathbb{R}^{n_j \times n_j}$ are the outputs of the singular value decomposition of matrix $\mathbf{M}_t^{n_j}$, and $\text{diag}(\cdot)$ returns a vector of the matrix main diagonal.

**3) Gradually injecting structure-aware regularization.** This strategy delicately calibrates the focus on patches throughout different learning steps, thereby ensuring a nuanced, stepwise infusion of structure-aware regularization. This judicious approach not only preserves the heterogeneity of the reconstructed samples but also optimizes the noise suppression process, culminating in a notable improvement in the overall performance of our model. Specifically, the modified regularization function $\mathcal{L}_s^t$ is written as

$$\mathcal{L}_s^t = \kappa_t \mathcal{L}_s(\mathbf{M}_{t-1}^{n_j}, \mathbf{M}_0^{n_j}), \ \kappa_t \sim \{\bar{\alpha_1}^2, \bar{\alpha_2}^2, ..., \bar{\alpha_t}^2\}, \ t \sim \{1, 2, ..., T\}, \tag{4}$$

where $\kappa_t$ denotes an adaptive factor for regularization term $\mathcal{L}_s(\mathbf{M}_{t-1}^{n_j}, \mathbf{M}_0^{n_j})$. This schedule allows for a more adaptive and flexible approach to regularization, enabling the model to capture better the global structures and details within the low-light images.

*Analysis of choosing rank-based global structure-aware regularization*. To minimize the curvature of ODE-trajectory, we bridge the deviation between the learnable closed-form trajectory samples $X_{t-1}$ and the GT $X_0$ by our rank-based global structure-aware regularization. Moreover, it is essential to acknowledge that while conventional pixel-wise regularization terms, including $L_1$, $L_2$, and SSIM, do offer some degree of enhancement to the original diffusion model, their impact is constrained. This limitation arises from their inadequate ability to fully encapsulate the non-local structures and intricate patterns within an image. Although regularization within the feature domain potentially aids in modeling a given image's structure, such structure is usually confined within a local region due to the kernel perceptual regions. This type of regularization, moreover, cannot explicitly characterize the properties of the structure and is lacking in theoretical guidance. Another significant concern is that regularization features could result in significant fluctuations in the back-propagated gradients, thereby impeding network training. Lastly, the empirical evidence documented in Table 4 corroborates the superiority of our rank-based global structure-aware regularization.

### 3.3 Integrating Uncertainty into Diffusion Process

To integrate uncertainty into the diffusion process, inspired by [25], we introduce an additional convolutional branch after the last layer of U-Net of the diffusion model for generating the uncertainty map. Specifically, we train the uncertainty model $\mu_{\theta_1}(\cdot)$ with the same step as the denoising diffusion models, like Algorithm 1, except that we optimize the following objective:

$$\mathcal{L}_\mu = \|e^{-\mathbf{P}_t} \odot (\, \epsilon - \epsilon_\theta(\mathbf{Y}, \mathbf{X}_t, \bar{\alpha}_t)\,)\|_1 + 2\,\mathbf{P}_t, \tag{5}$$

where $\mathbf{P}_t := \mu_{\theta_1}(\mathbf{Y}, \mathbf{X}_t, \bar{\alpha}_t)$ denotes a pixel-wise uncertainty map, i.e., $\mathbf{P}_t \in \mathbb{R}^{H \times W \times 3}$, $\epsilon_\theta(\cdot)$ represents the corresponding function with weights of a denoising diffusion model. Upon completion of the uncertainty estimation model's pretraining phase, we obtain $\mu_{\hat{\theta}_1}$, and $\epsilon_{\hat{\theta}}$. Subsequently, we fix the parameter of $\mu_{\hat{\theta}_1}$ to generate the uncertainty map $\mathbf{P}_t$. This map serves as a flexible weight, calibrating the importance of each pixel such that pixels with higher uncertainty are assigned greater weight. Further details are provided in Algorithm 3. The integration of uncertainty into the diffusion process in this manner incentivizes the diffusion model to intensify its focus on uncertain or challenging regions. This strategic approach yields visually pleasing results, underscoring the value of incorporating uncertainty-aware regularization in the enhancement of low-light images.

---

**Algorithm 3** Training a diffusion-based low-light enhancement model

---

**Require:** Normal and low-light image pairs $(\mathbf{X}_0, \mathbf{Y})$, and a pre-trained models $\mu_{\theta_1}$ and $\epsilon_{\hat{\theta}}$.

1: Initialize parameters of $\mu_{\theta_1}(\cdot)$ (resp. $\epsilon_\theta(\cdot)$) from pretrained $\hat{\theta}_1$ (resp. $\hat{\theta}$), and freeze $\hat{\theta}_1$.
2: **Repeat**
3:     Sample an image pair $(\mathbf{X}_0, \mathbf{Y})$, $\bar{\alpha}_t \sim p(\bar{\alpha})$ and $\epsilon \sim \mathcal{N}(0, \mathbf{I})$
4:     $\mathbf{X}_t = \sqrt{\bar{\alpha}_t}\,\mathbf{X}_0 + \sqrt{1 - \bar{\alpha}_t}\,\epsilon$
5:     $\mathbf{P}_t \leftarrow \mu_{\theta_1}(\mathbf{Y}, \mathbf{X}_t, \bar{\alpha}_t)$
6:     Construct the learnable closed-form sample $\mathbf{X}_{t-1}$ via Eq. (2)
7:     Perform a gradient descent step on
8:     $\nabla_\theta\{\lambda\|\mathbf{P}_t \odot \epsilon - \mathbf{P}_t \odot \epsilon_\theta(\mathbf{Y}, \mathbf{X}_t, \bar{\alpha}_t)\|_1 + \mathcal{L}_s^t\,\}$, where $\lambda$ is a balancing factor, and $\mathcal{L}_s^t$ refers to Eq. (4)
9: **Until** converged
10: **Return** the resulting diffusion model $\epsilon_{\tilde{\theta}}$

---

## 4 Experiments

### 4.1 Experiment Settings

**Datasets.** We employed seven commonly-used LLIE benchmark datasets for evaluation, including LOLv1 [45], LOLv2 [54], DICM [11], LIME [6], MEF [14], NPE [39], and VV[2]. Specifically, LOLv1 contains 485 low/normal-light image pairs for training and 15 pairs for testing, captured at various exposure times from the real scene. LOLv2 is split into two subsets: LOLv2-real and LOLv2-synthetic. LOLv2-real comprises 689 pairs of low-/normal-light images for training and 100 pairs for testing, collected by adjusting the exposure time and ISO. LOLv2-synthetic was generated by analyzing the illumination distribution of low-light images, consisting of 900 paired images for training and 100 pairs for testing. In accordance with the settings outlined in the recent works [51, 43, 40], we trained and tested our models on LOLv1 and LOLv2 datasets separately. The DICM, LIME, MEF, NPE, and VV datasets contain several unpaired real low-light images, which are used only for testing.

**Evaluation metrics.** We adopted both full-reference and non-reference image quality evaluation metrics to evaluate various LLIE approaches. For paired data testing, we utilized peak signal-to-noise ratio (PSNR), structural similarity (SSIM) [44], and learned perceptual image patch similarity (LPIPS) [57]. For datasets such as DICM, LIME, MEF, NPE, and VV, where paired data are unavailable, we adopted only the naturalness image quality evaluator (NIQE) [20].

**Methods under comparison.** We compared our method with a great variety of state-of-the-art LLIE methods, i.e., LIME [6], RetinexNet [45], KinD [60], Zero-DCE [5], DRBN [52], KinD++ [59],

---

[2]https://sites.google.com/site/vonikakis/datasets

Table 1: Quantitative comparisons of different methods on LOLv1 and LOLv2. The best and second-best results are highlighted in **bold** and underlined, respectively."↑" (resp. "↓") means the larger (resp. smaller), the better. Note that we obtained these results either from the corresponding papers, or by running the pre-trained models released by the authors, and some of them lack relevant results on the LOLv2-synthetic dataset.

| Methods | LOLv1 | | | LOLv2-real | | | LOLv2-synthetic | | | Param (M) |
|---|---|---|---|---|---|---|---|---|---|---|
| | PSNR↑ | SSIM↑ | LPIPS↓ | PSNR↑ | SSIM↑ | LPIPS↓ | PSNR↑ | SSIM↑ | LPIPS↓ | |
| LIME [6] TIP'16 | 16.760 | 0.560 | 0.350 | 15.240 | 0.470 | 0.415 | 16.880 | 0.776 | 0.675 | - |
| Zero-DCE [5] CVPR'20 | 14.861 | 0.562 | 0.335 | 18.059 | 0.580 | 0.313 | - | - | - | 0.33 |
| EnlightenGAN [8] TIP'21 | 17.483 | 0.652 | 0.322 | 18.640 | 0.677 | 0.309 | 16.570 | 0.734 | - | 8.64 |
| RetinexNet [45] BMVC'18 | 16.770 | 0.462 | 0.474 | 18.371 | 0.723 | 0.365 | 17.130 | 0.798 | 0.754 | 0.62 |
| DRBN [52] CVPR'20 | 19.860 | 0.834 | 0.155 | 20.130 | 0.830 | 0.147 | 23.220 | 0.927 | - | 2.21 |
| KinD [60] MM'19 | 20.870 | 0.799 | 0.207 | 17.544 | 0.669 | 0.375 | 16.259 | 0.591 | 0.435 | 8.03 |
| KinD++ [59] IJCV'20 | 21.300 | 0.823 | 0.175 | 19.087 | 0.817 | 0.180 | - | - | - | 9.63 |
| MIRNet [55] TPAMI'22 | 24.140 | 0.842 | 0.131 | 20.357 | 0.782 | 0.317 | 21.940 | 0.846 | - | 5.90 |
| LLFlow [43] AAAI'22 | 25.132 | 0.872 | 0.117 | 26.200 | 0.888 | 0.137 | 24.807 | 0.9193 | 0.067 | 37.68 |
| LLFormer [40] AAAI'23 | 25.758 | 0.823 | 0.167 | 26.197 | 0.819 | 0.209 | 28.006 | 0.927 | 0.061 | 24.55 |
| SNR-Aware [51] CVPR'22 | 26.716 | 0.851 | 0.152 | 27.209 | 0.871 | 0.157 | 27.787 | 0.941 | 0.054 | 39.13 |
| Ours | **27.839** | **0.877** | **0.091** | **28.818** | **0.895** | **0.095** | **28.670** | **0.944** | **0.047** | 17.36 |

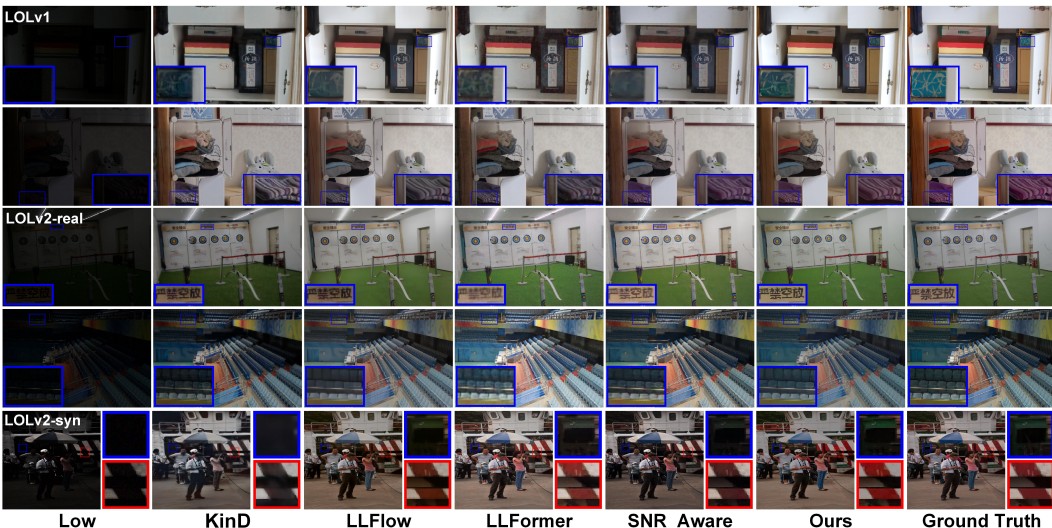

Figure 3: Visual comparisons of the enhanced results by different methods on LOLv1 and LOLv2.

EnlightenGAN [8], MIRNet [55], LLFlow [43], DCC-Net [61], SNR-Aware [51], and LLFormer [40].

**Implementation details.** We trained our models for 2M iteration steps with PyTorch. We employed an Adam optimizer [10] with a fixed learning rate of $1 \times 10^{-4}$ without weight decay. We applied an exponential moving average with a weight of 0.9999 during parameter updates. During training, we utilized a fine-grained diffusion process with $T = 500$ steps and a linear noise schedule with the two endpoints of $1 \times 10^{-4}$ and $2 \times 10^{-2}$. The patch size and batch size were set to 96 and 8, respectively. The hyper-parameter $\lambda$ was empirically set to 10.

## 4.2 Comparison with State-of-the-Art Methods

**Results on LOLv1 and LOLv2.** Table 1 shows the quantitative results of different LLIE methods, where it can be observed that our method consistently achieves the best performance over all the compared methods in terms of PSNR, SSIM, and LPIPS metrics. Particularly, the remarkable improvement on **LPIPS** provides compelling evidence of the superior perceptual quality of our method, as shown in Fig. 3, where our method effectively suppresses artifacts and reveals image details, leading to visually appealing results that are more faithful to the original scene.

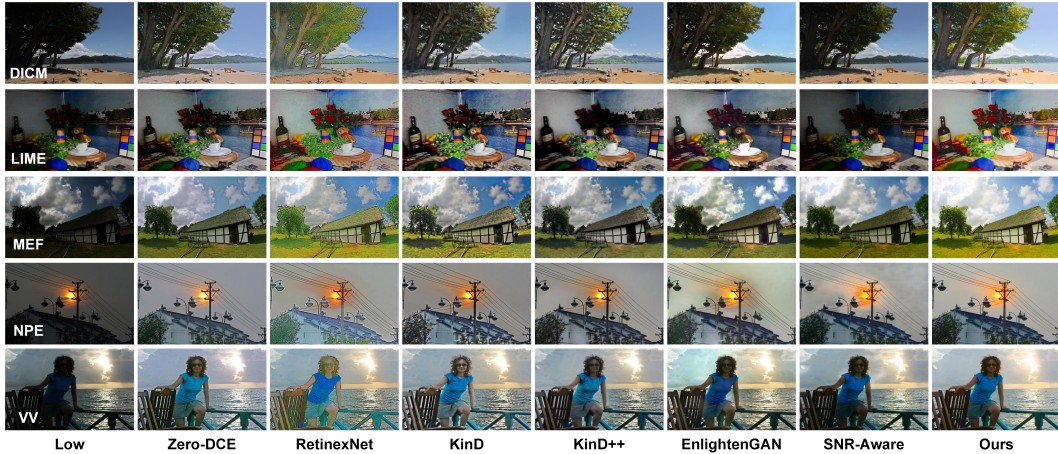

Figure 4: Visual comparisons of different methods on DICM, LIME, MEF, NPE, and VV datasets.

**Results on DICM, LIME, MEF, NPE, and VV.** We directly applied the model trained on the LOLv2-synthetic dataset to these unpaired real-world datasets. From Table 2, we can observe that our method yields better NIQE scores than all the other competitors, demonstrating its stronger generalization ability. In addition, a visual comparison of enhancement results from various methods, as showcased in Fig. 4, underscores the superior performance of our approach. Evidently, our model adeptly adjusts the illumination conditions, optimally enhancing the visibility in low-light areas while vigilantly circumventing over-exposure. This successful balance corroborates the superior efficacy of our proposed method.

Table 2: Quantitative comparisons of different methods on DICM, LIME, MEF, NPE, and VV datasets in terms of NIQE, where smaller values lead to better performance.

| Methods | DICM | LIME | MEF | NPE | VV |
|---|---|---|---|---|---|
| Zero-DCE [5] | 4.58 | 5.82 | 4.93 | 4.53 | 4.81 |
| EnlightenGAN [8] | 4.06 | 4.59 | 4.70 | 3.99 | 4.04 |
| RetinexNet [45] | 4.33 | 5.75 | 4.93 | 4.95 | 4.32 |
| KinD [60] | 3.95 | 4.42 | 4.45 | 3.92 | 3.72 |
| KinD++ [59] | 3.89 | 4.90 | 4.55 | 3.91 | 3.82 |
| SNR-Aware [51] | 6.12 | 5.93 | 6.44 | 6.44 | 11.50 |
| DCC-Net [61] | 3.70 | 4.42 | 4.59 | 3.70 | 3.28 |
| Ours | **3.28** | **4.32** | **3.40** | **3.55** | **2.60** |

Table 3: Ablation studies of effectiveness of various clustering algorithms used in our method.

| Clustering Algorithms | Baseline | K-Means | Spectral clustering | Gaussian Mixture Model | Hierarchical clustering |
|---|---|---|---|---|---|
| PSNR↑ | 26.02 | 27.02 | 27.41 | 27.55 | 27.70 |
| SSIM↑ | 0.859 | 0.872 | 0.876 | 0.877 | 0.880 |
| LPIPS↓ | 0.123 | 0.097 | 0.095 | 0.095 | 0.092 |

Table 4: Ablation studies of the effectiveness of choosing our rank-based global structure-aware regularization.

| Regularization | Baseline | $L_1$ loss | $L_2$ loss | SSIM | Perceptual loss | Our rank-based modeling |
|---|---|---|---|---|---|---|
| PSNR↑ | 26.02 | 26.45 | 26.53 | 26.23 | 26.42 | 27.70 |
| SSIM↑ | 0.859 | 0.868 | 0.869 | 0.870 | 0.863 | 0.880 |
| LPIPS↓ | 0.123 | 0.101 | 0.102 | 0.101 | 0.099 | 0.092 |

## 4.3 Ablation Study

**Clustering algorithms.** We investigated how the clustering method used for finding similar patches affects the quantitative performance on LOLv1. As shown in Table 3, our method generally benefits from different clustering algorithms. Especially, when employing advanced hierarchical clustering, the PSNR value experiences an improvement of approximately 0.7 dB, compared to the results achieved with the K-Means clustering. This further underscores the prospective utility of our

Table 5: Quantitative results of the ablation studies on LOLv1. "✗" (resp. "✓") means the corresponding module is unused (resp. used). (a) Non-local patch-based matrix rank modeling, (b) Gradually injecting structure-aware regularization, and (c) Uncertainty-guided regularization. Note that (b) is applicable only if (a) is utilized.

|  | (a) | (b) | (c) | PSNR↑ | SSIM↑ | LPIPS↓ |
|---|---|---|---|---|---|---|
| 1) | ✗ | ✗ | ✗ | 26.02 | 0.8593 | 0.1226 |
| 2) | ✗ | ✗ | ✓ | 26.79 | 0.8708 | 0.1049 |
| 3) | ✓ | ✗ | ✗ | 27.02 | 0.8753 | 0.0980 |
| 4) | ✓ | ✓ | ✗ | 27.70 | 0.8795 | 0.0923 |
| 5) | ✓ | ✓ | ✓ | 27.84 | 0.8774 | 0.0905 |

Figure 5: Visual results of the four settings of the ablation studies in Table 5.

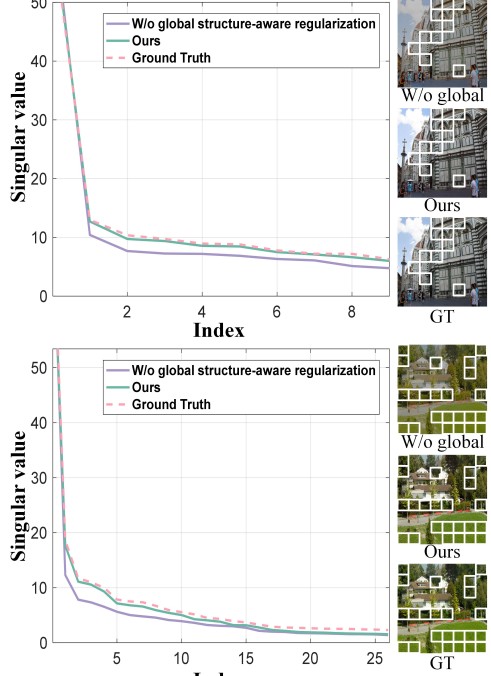

regularization term in enhancing diffusion models with advanced clustering algorithms. Consequently, we adopted advanced hierarchical clustering in our approach for all subsequent experiments.

**Regularization terms between $X_{t-1}$ and $X_0$.** We delved into the impact of utilizing varying regularization terms between $X_{t-1}$ and $X_0$ on the enhancement of diffusion models within LOLv1. Table 4 offers insights that although conventional pixel-wise regularization terms such as $L_1$, $L_2$, and SSIM can moderately elevate the performance of the baseline, their enhancements pale in comparison to our rank-based modeling regularization. The primary reason is the inability of these regularization terms to fully encapsulate the nonlocal structures or patterns inherent within an image. These outcomes distinctly highlight the superiority of our rank-based model in capturing global structures.

**Non-local patch-based matrix rank modeling & adaptive regularization schedule.** We established a baseline by demounting the non-local matrix rank-based representation learning regularization. By comparing rows 2), 3), 4), and 5) in Table 5, it can be observed that this module equipped with our adaptive regularization schedule significantly boosts the enhancement performance. However, this module without adopting the adaptive regularization schedule only achieves minor improvement. Such observations demonstrate the necessity of our adaptive regularization schedule and the effectiveness of non-local rank-based representation learning.

Furthermore, as depicted in Fig. 6, we elaborate on the matrix rank across clusters from several samples, thereby offering a clear visualization of the effectiveness of this regularization. Empirical findings affirm that implementation of our proposed regularization notably narrows the gap of singular values between the reconstruction output and the ground truth, thereby substantiating the indispensability of this regularization approach and validating its strategic integration into the diffusion process.

Figure 6: Visual comparisons of the distribution of the singular values of a cluster from enhanced images by our method with and without global structure-aware regularization. Note that the patches contained in the cluster are highlighted with a white box.

**Uncertainty-guided regularization.** The effectiveness of this regularization is substantiated through a comparative analysis of rows 1) and 2) of Table 5. Notably, the implementation of this regularization is evidenced to bolster the performance of image enhancement. Such improvement, however, is confined to pixel-wise metrics, e.g., PSNR and SSIM, and does not extend to global visual metrics

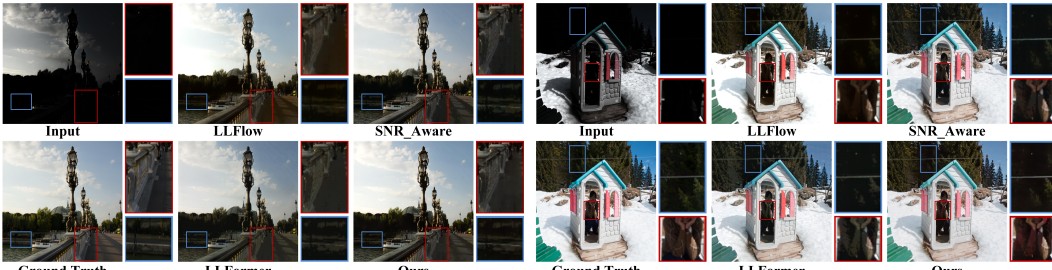

Figure 7: Illustration of challenging scenarios, where both the proposed method and other existing methods exhibit limitations in achieving visually pleasing reconstructions.

such as LPIPS (visualization as Fig. 5). This observation implies that the incorporation of uncertainty-based regularization in isolation yields incremental advancements but fails to induce substantial modifications to the structure of the reconstructed images. The findings, therefore, underscore the necessity of an integrated approach that combines uncertainty-based regularization with global structure-aware regularization to achieve significant improvements in both local pixel-level and global structural aspects of image enhancement.

### 4.4 Challenging Cases

While the proposed method has demonstrated notable advancements in performance, as showcased in the previous sections, it is important to acknowledge that certain challenging scenarios persist within the domain of low-light enhancement tasks. Fig. 7 illustrates two challenging examples that highlight the intricacy of the task at hand. In these examples, it is evident that certain regions of the input image are under extreme low-light conditions, to the point where any trace of texture becomes indiscernible. This lack of visual information poses a significant challenge, causing both our proposed method and other state-of-the-art techniques to struggle in generating visually compelling details. To tackle these challenges, we could identify a subset of extremely low-light patches and apply robust regularization to them to improve reconstruction performance.

## 5 Conclusion and Discussion

We have introduced a diffusion model-based approach for enhancing low-light images effectively. The cornerstone of our approach lies in the innovative training strategy of diffusion models from the perspective of minimizing the ODE-trajectory curvature, which involves the global structure-aware rank-based regularization on learnable closed-form samples and uncertainty-based learning of latent noises. Recognizing the potential adverse effects of regularization within the early stage of diffusion models, we have thoughtfully devised a progressively adaptive regularization schedule. This strategy ensures a nuanced, step-wise infusion of structure-aware regularization, preserving the inherent structural integrity in the reconstructed samples. As we continue to refine our approach, we anticipate further improvements and look forward to the potential applications and advancements these developments will facilitate in broader fields.

While our work exhibits admirable performance compared with contemporary methods, we acknowledge the inherent computational burden associated with diffusion-based methods—attributable to their iterative noise removal in a reverse Markov chain. Nonetheless, thanks to the latest strides in the field [33, 35], it is anticipated that the diffusion models may soon be capable of delivering accelerated reconstructions through a minimal number of iterative steps.

## Acknowledgements

This work was supported in part by the Hong Kong Research Grants Council under Grant CityU 11218121, in part by the Hong Kong Innovation and Technology Fund under Grant MHP/117/21, and in part by Hong Kong University Grants Committee under Grant UGC/FDS11/E02/22, and in part by the Natural Science Foundation of Shandong Province under Grant ZR2022ZD38.

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
