# OpenReview forum: "Global Structure-Aware Diffusion Process for Low-light Image Enhancement"
_NeurIPS.cc/2023/Conference — NeurIPS 2023 poster_

### Official Review · Reviewer_2x92 · 2023-06-28

**Soundness:** 3 good
**Presentation:** 3 good
**Contribution:** 2 fair
**Rating:** 3
**Confidence:** 5

**Summary:**

This work introduces a low-light image enhancement based on diffusion model. By incorporating global structure-aware regularization, this work achieves more promising performances than existing methods.

**Strengths:**

By considering the global structure during the diffusion process, this work achieves better performance.

**Weaknesses:**

Despite its better performance, I am very sure that this work cannot advance this field. It does not solve a problem in a new scenario and it just make incrimental revision on top of established framework. My concerns for this work can be concluded as follows:
1. The authors claim that a naive implementation of the diffusion model alone is inadequate for effectively resolving the low-light image enhancement, while it does not prove this claim. All baselines considered in the experiments are non-diffusion based approaches. Thus, it is difficult to say whether a naive implementation is enough for the problem solving.
2. In line 151, the autors claim that the rank of matrix is an effective measurement of the global structures. This is a very important assumptions in this place. This part needs more experiments and justifactions. Or, the authors should provide very solid references for this claim in this place.
3. Except the matrix ranking, several approachs have been proposed to better represent the global structures including some deep learning features. The authors should compare with them.
4. Why is it important to use global structure in this place? If U-Net is a substantial structure of this proposed framework, it is able to capture the global structure during its network propagation. Thus, why do this work design an additional module to capture global structure.
5. I am not sure about the effectiveness of uncertainty in this place. I cannot find Table 5 mentioned in this manuscript.
6. This work just puts several different image processing modules together without enough insights. It is easy to come up with such an idea. The non-local solutions have been discussed in traditional solutions. I donot know why they choose to combine it with diffusion model. Besides, weighted loss has been considered for a large number of methods. Using it here cannot be considered as a new contribution.

**Questions:**

See my question above.

**Limitations:**

See my questions above.

---

> ### Author Rebuttal · Authors · 2023-08-09
>
> ### **[Responses to Reviewer 2x92]**
>
> ### ***1** Response to Weakness 1 (W1): Baseline Model*
>
> We refer the reviewer to the results of ablation studies depicted in Figure 5 on page 9, where the first case corresponds to the baseline diffusion model. It only achieves
> 26.02 dB, which is noticeably lower than the SOTA method SNR-Aware's achievement of 26.70 dB.
>
> ### ***2** Response to W2: Theoretical Soundness of Rank-based Regularization*
>
> First, we clarify that as stated in Lines 112 - 114, the ``global structure" means that the non-local similar patterns.  We have indeed cited some related works [5, 48] in Sec 3.1.1  of the manuscript. Moreover, [41 , 16] also acknowledge the effectiveness of a matrix rank to model the image global structure. Besides, it is somewhat common sense that the rank of a 2D matrix measures the correlation between its elements in a global manner rather than an element-wise manner.
>
> ### ***3** Response to W3: Different loss Terms*
>
> We made comparisons with the perceptual loss, i.e., we replaced our rank-based regularization with the perceptual loss. The quantitative results listed in the following table show ours outperforms the perceptual loss to a large extent. We also refer the reviewer to **the second response to Reviewer gyFW** for the results of more loss terms.
>
> | Methods                  | PSNR      | SSIM      | LPIPS     |
> |:--------------------------:|-----------|-----------|:----------|
> | Baseline                 | 26.02     | 0.859     | 0.123     |
> | Perceptual loss          | 26.42     | 0.863     | 0.099     |
> | Our matrix rank modeling | **27.02** | **0.872** | **0.097** |
>
> ### ***4** Response to W4: Architecture or Loss Terms for Structure-aware Diffusion Process*
>
> We note numerous factors influence network performance, where the network structure represents merely one of these variables. A network may not reach its full potential without the appropriate configuration of these diverse training settings. While the U-Net architecture is adept at integrating feature maps across multiple resolutions, the vanilla loss doesn't thoroughly harness these properties. We also refer the reviewer to the **second response to Reviewer gyFW** for the detailed quantitative and theoretical analyses of introducing adaptive rank-based regularization into the diffusion model.
>
> For your convenience, we have pasted some of the content.
>
> " .....
> While the current approaches often involve pixel-wise treatments, inherent global structures can be overlooked to some extent. Modeling such global structures potentially augments the performance [C3]. Additionally, conventional pixel-wise regularization terms, such as L1, L2, and SSIM, do not adequately encapsulate nonlocal structures. However, regularization within the feature domain is usually confined within a local region due to the kernel perceptual regions.  Moreover, regularizing features can result in fluctuations in the back-propagated gradients, thereby impeding network training.
>
> Finally, the results in the following table also verify the above analyses, i.e., the necessity of introducing regularization between $X_{t-1}$ and $X_0$, and the advantage of our rank-based model.
>
> |Method| PSNR| SSIM| LPIPS|
> |--|--|--|--|
> | Baseline| 26.02| 0.859| 0.123 |
> | L1 Reg. | 26.45| 0.868| 0.101 |
> | L2 Reg. | 26.53| 0.869| 0.102 |
> | SSIM Reg.| 26.23| 0.870| 0.101|
> | Perceptual Reg.| 26.42| 0.863| 0.099|
> | Rank Reg. (Ours) | **27.02** | **0.872** | **0.097** |
>
> [C3] S. Gu, et al., Weighted nuclear norm minimization with application to image denoising, CVPR'14.
> "
>
>
> ### ***5** Response to W5: Missing of Table 5*
>
> Sorry for our mistake.  Current Figure 5 indicates the ablation studies of different loss terms. We will correct the label in the final version.
>
> ### ***6** Response to W6: Contributions of the Method*
>
> **We beg to differ in your evaluation of our method. Our method is by no means an assembly of prior art.** Our major contribution lies in introducing global structure-aware regularization into the learning process of the diffusion model, which **Reviewer U5ii** acknowledges. Current diffusion models, with their naive loss terms, cannot fully capture the global properties. Thus, we introduce non-local patch-based matrix rank modeling to address such an issue. Moreover, our regularization method tends to minimize the trajectory curvature, which potentially helps image reconstruction [C1,C2] and leads to outstanding performance compared to SOTA methods. We refer the reviewer to the **second response to Reviewer gyFW** for the detailed analyses of the necessity and advancement of introducing rank-based regularization into the diffusion process for capturing the global structure and the theoretical analysis in the attached pdf file.
>
> Besides, it is also worth mentioning that, as discussed in the **first response to Reviewer U5ii** , the performance of our method is further improved significantly by applying advanced clustering methods for grouping non-local patches, i.e., the overall PSNR value increases about 0.7 dB, compared to those reported in our manuscript.
>
> We posit that our contributions will pave the way for fresh perspectives on diffusion models for low-level image processing tasks. And our method improves the SOTA performance of low-light image enhancement to a higher level, which will contribute to the community.
>
> [C1] S. Lee, et al., Minimizing trajectory curvature of ode-based generative models, ICML'23.
>
> [C2] X. Liu, et al., Flow straight and fast: Learning to generate and transfer data with rectified flow,  ICLR'23.

---

> > ### Comment · Reviewer_2x92 · 2023-08-14
> > **Global structure**
> >
> > Thanks for the rebuttal. If the global structure is an important contribution, have you ever done more experiments to validate the effectiveness of your proposed framework with other established ones.

---

> > > ### Author Response · Authors · 2023-08-14
> > > **Comparison with other diffusion-based low-light image enhancement methods**
> > >
> > > We are pleased to hear back from the reviewer. Regarding the comparison with “other established ones,” we assume the reviewer is referring to the comparisons with other diffusion-based low-light image enhancement methods. Actually, we indeed made it. We refer the reviewer to the $4^{th}$ response to **Reviewer ts6H**, where we compared our method with the pyramid diffusion [C3], the latest diffusion-based method for low-light enhancement published in IJCAI’23. For your convenience, we have pasted some results here.
> > >
> > > | Methods    | Architecture    | Loss term                                                 | PSNR   | SSIM  | LPIPS |
> > > |------------|---------------|------------------------------------------|------|-----|----|
> > > | PyDiff [C3] | Pyramid         | Vanilla with Multi-scale L1                               | 27.090 | 0.879 | 0.100 |
> > > | Ours       | Vanilla (U-Net) | Rank-based modeling with basic KMeans clustering          | 27.336 | 0.874 | 0.097 |
> > > | Ours++     | Vanilla (U-Net) | Rank-based modeling with advanced Hierarchical clustering | **27.697** | **0.880** | **0.092** |
> > >
> > > Additionally, when comparing other augmentation strategies or loss terms, we've evaluated our method's performance against L1, L2, and Perceptual losses. Please see our 2nd response to **Reviewer gyFW** for further comparison details.
> > >
> > > |Method| PSNR| SSIM| LPIPS|
> > > |--|--|--|--|
> > > | Baseline| 26.02| 0.859| 0.123 |
> > > | L1 Reg. | 26.45| 0.868| 0.101 |
> > > | L2 Reg. | 26.53| 0.869| 0.102 |
> > > | SSIM Reg.| 26.23| 0.870| 0.101|
> > > | Perceptual Reg.| 26.42| 0.863| 0.099|
> > > | Rank Reg. (Ours) | **27.02** | **0.872** | **0.097** |
> > >
> > > Last but not least, we also want to note that regarding our main contributions,  the motivation/findings of introducing regularization between $X_{t-1}$ and $X_{0}$ is also essential , as clarified in the $2^{nd}$ response to **Reviewer gyFW**
> > >
> > > [C3] Zhou, Dewei, et al. Pyramid Diffusion Models For Low-light Image Enhancement, IJCAI'23.

---

> > > > ### Comment · Reviewer_2x92 · 2023-08-18
> > > > **My feedback**
> > > >
> > > > Thanks for the feedback. Diffusion has become mainstream for low-level vision. Does this mean low-light image enhancement based on diffusion is an easy idea to come up with? For any new papers on low-light image enhancement, we should set very high standards. In my opinion, this work is too generic. Such a method can be easily employed for other low-level vision problems like underwater image enhancement, image dehazing, or even denoising. We may be unable to find enough specific low-light image enhancement methods in this situation. It is better to say that this is a framework designed for general low-level vision problems, since global structure is also useful for other tasks. In this regard, the authors are suggested to conduct more experiments on aforementioned areas to investigate whether this proposed method is applicable.

---

> > > > > ### Author Response · Authors · 2023-08-19
> > > > >
> > > > > Thanks again for your feedback. Based on the comments, it appears that we have addressed all technical issues you posed previously. Here's our detailed rebuttal to your queries regarding our contributions:
> > > > >
> > > > > We agree with you that introducing diffusion-based methods into low-light enhancement is **somewhat** easy to come up with. Our selection of this subject stems from our ongoing research project's needs and our sustained efforts in this area. Several **concurrent** works on diffusion-based low-light image enhancement have been proposed, such as Diffusion-prior in CVPR’23 [C1] and Pyramid diffusion in IJCAI’23 [C2].  They primarily emphasize on crafting novel architectures, yet don’t yield impressive performance. Thus, **how to adapt the diffusion process effectively and efficiently to low-light image enhancement remains a tough issue**. **By contrast**, we innovatively model the latent global structure in a low curvature trajectory reverse process (see the detailed clarification in the second response to **Reviewer gyFW**) that is strongly recognized by **Reviewer U5ii**, instead of simply introducing a vanilla diffusion model. As experimentally demonstrated, our method with the advanced patch grouping algorithm improves the SOTA work (SNR-aware) with 1 dB, and **significantly** suppresses the concurrent work Pyramid diffusion [C2] IJCAI’ 23 (see the 4th response to **Reviewer ts6H**).
> > > > >
> > > > >
> > > > > Second, we recognize that there are two distinct research guidelines: one that offers diverse yet suboptimal performance, and another that focuses on a specific domain yielding admirable results. **Both paradigms are meaningful and necessary**. We believe **it's unfair to underestimate the significance of our novel technique and notable improvement (by over 1 dB) just because its potential applications in other areas haven't been explored**. Moreover, we have fully exploited the potential of diffusion models for low-light enhancement using additional plug-and-play regularization term. **This, in itself, stands as a complete work.**
> > > > >
> > > > > Last, as we delved deeper into this work, we became increasingly aware that our theoretical insights and empirical findings of the plug-and-play regularization to the diffusion process may benefit various diffusion-based low-level image restoration/reconstruction tasks. However, if we were to achieve the same magnitude of performance improvement in other areas as we did in low-light enhancement, we believe the workload would surpass that of a single conference submission. But, as evidence of the generality of our method, we have **indeed** conducted super-resolution experiments in the first response to **Reviewer 9ew6**. We also copy the results here for your convenience.
> > > > > | Methods                                        | PSNR  | SSIM  |
> > > > > |------------------------------------------------|-------|-------|
> > > > > | IDM [C3]                                       | 24.01 | 0.710 |
> > > > > | Ours w/o global structure-aware regularization | 23.94 | 0.713 |
> > > > > | Ours                                           | 24.52 | 0.728 |
> > > > >
> > > > > [C1]B. Fei, et al., Generative Diffusion Prior for Unified Image Restoration and Enhancement, CVPR'23.
> > > > >
> > > > > [C2]D., Zhou, et al. Pyramid Diffusion Models For Low-light Image Enhancement, IJCAI'23.
> > > > >
> > > > > [C3]S. Gao, et al., Implicit diffusion models for continuous super-resolution, CVPR'23.

---

> > > > > > ### Comment · Reviewer_2x92 · 2023-08-19
> > > > > > **My feedback**
> > > > > >
> > > > > > Please do not make any assumptions on my behalf. I am bringing up this concern because I believe it is more significant as it pertains to the motivation behind this work. This does not imply that my other concerns have been resolved.
> > > > > >
> > > > > > Second, it would be unjust to claim that other works lack promising performance. As a highly competitive research field, I do not believe that their papers would be accepted by reviewers if they did not demonstrate promising performance. Since some ICCV papers has been released on arxiv, we can find more low-ligh image enhancement papers on diffusion (ignore this sentence, since ICCV is still on its way.)
> > > > > >
> > > > > > Third, it is important to acknowledge the previous works that have shown remarkable and promising performance in utilizing diffusion for low-light image enhancement. It is unclear whether you have reviewed these papers during the preparation. Discovering the potential of established structures in new areas is crucial for the advancement of a field. If you are publishing a paper on top of these works, it is essential to have significant results to demonstrate. Otherwise, what would be the purpose of publishing the article?
> > > > > >
> > > > > > Upon reviewing your response to other reviewers regarding the non-local patch, I have a question. I recall that the concept of utilizing non-local priors has been extensively applied in various image restoration problems, even in the use of convex optimization and sparse coding. What are the primary advantages of your approach in comparison to previous methods in this area? At that time, non-local prior is also a regularization term in covex optimization equation, just like the loss functions in this place.
> > > > > >
> > > > > > Moreover, I do not make any judgements about whether your architeture is novel or not. Besides the novelty, I just say that I feel this structure is rather generic..
> > > > > >
> > > > > > At last, let us say something beyond this paper. From the usage of the large-scale models, it seems that "one that offers diverse yet suboptimal performance" may be a mainstream in the future.

---

> > > > > > > ### Author Response · Authors · 2023-08-20
> > > > > > >
> > > > > > > Thanks for the additional feedback, we will try to resolve all your concerns.
> > > > > > >
> > > > > > > **Performance of SOTA diffusion methods**. We want to claim that there are still potential for applying diffusion-based network to low-light enhancement. At their submission stage, their performance was indeed promising and admirable. However, with time passing, the requirement for high performance is gradually increasing. Especially, we have found that applying global structure regularization could greatly improve diffusion model performance. Moreover, our method is **the best method among all the compared methods**, we acknowledged, upon the most widely adopted datasets.
> > > > > > >
> > > > > > > **Other SOTA methods**. We have made our best to find SOTA methods, e.g., LLFlow AAAI'22, LLFormer AAAI'23, and SNR-Aware CVPR' 22, which can representatively show the state-of-the-art methods. There may be some papers that we have overlooked, but it is impractical for a single paper to review all published related work.
> > > > > > >
> > > > > > > **Advantage of using global structure-regularization**. Our contribution/ advantage mainly lies in introducing the global-structure modeling process into diffusion models.  We have developed a method to **seamlessly integrate non-local priors into the diffusion model**, which results in superior performance. While there are indeed various methods to embed non-local priors, such as the optimization-based methods you discussed, these cannot be adaptively incorporated into the diffusion model, or at least, **no existing literature indicates this**. Moreover, in the realm of image reconstruction, there are currently no works attempting to merge classic image priors into diffusion models.
> > > > > > >
> > > > > > > **A generic method**. Yes, our method could be generally applied to other image reconstruction problems. We have validated their superior performance in low-light enhancement and super-resolution. We firmly believe that our contribution is substantial enough for a conference submission, which also is strongly recognized by **Reviewer U5ii**.  Furthermore, we will also validate their performance in other related fields.
> > > > > > >
> > > > > > > **Large-scale models**.  We agree that large-scale models represent a promising approach. However, other specialized models also hold significant value in the computer vision community. Their practical usefulness in real-world applications cannot be overlooked.
> > > > > > >
> > > > > > > **Last but not least, we are pleased to address your remaining unsolved technical concerns to enhance the quality of the paper.**

---

> > > > > > > > ### Comment · Reviewer_2x92 · 2023-08-21
> > > > > > > > **Comments**
> > > > > > > >
> > > > > > > > Thanks for your paper.  I may need to consider my final rating at a later stage.
> > > > > > > >
> > > > > > > > You mention that your global structure-regularization is "seamlessly integrate," but it's just a loss function. Your specific design can just be found in Line 157. However, it only considers the step-by-step generation of the diffusion process. It is not an appropriate idea to call it "seamlessly". Lines 146 to 156, however, are not new to me. This part is actually very similar to a lot of classical papers published about eight years ago. Such methods have been attacked many times when deep learning dominates the society. Besides, please cite more papers in Line 146 to Line 156, since you are not the first to come up with such ideas. You cite those paper in Line 113, which may make readers feel that you propose those ideas alone..

---

> > > > > > > > > ### Author Response · Authors · 2023-08-21
> > > > > > > > > **Rebuttal by Authors**
> > > > > > > > >
> > > > > > > > > Thanks for the further feedback.
> > > > > > > > >
> > > > > > > > > **Seamless integration**. We wish to emphasize that integrating regularization terms into the training phase of diffusion models presents inherent challenges. During the training phase, the network's output is **just the latent noise map** instead of a clean image. This demands **careful consideration of multiple factors** for effective regularization. For instance, optimal placement of regularization between variables (whether between $\hat{X_0}$ and $X_0$ or between $\hat{X}_{t-1}$ with $X_0$, et al., where hat means the reconstructed one),  the potential necessity of introducing noise into the closed-form samples analogous to the reverse process, and the dynamic coordination of regularization weights. Although our findings are presented in a concise manner, they consist of our extensive research efforts. We've appended the relevant experimental results for your review.
> > > > > > > > >
> > > > > > > > > |Constraint items| PSNR | SSIM|
> > > > > > > > > |:-|-|-|
> > > > > > > > > | Baseline: W/O regularization  | 26.02 | 0.859 |
> > > > > > > > > | + Regularization between $\hat{X}_{0}$ and $X_0$  | 26.53 | 0.865 |
> > > > > > > > > | + Reg. btw. $\hat{X}_{t}$ and $X_0$ | 26.39 | 0.862 |
> > > > > > > > > | + Reg. btw. $\hat{X}_{t-1}$ without noise and $X_0$ | 26.61 | 0.867 |
> > > > > > > > > | ... | ...   | ...   |
> > > > > > > > > | + Ours: $\hat{X}_{t-1}$ and $X_0$   | 27.02 | 0.872 |
> > > > > > > > >
> > > > > > > > > In addition, we've provided a theoretical analysis of our regularization term in response to **Reviewer gyFW**. For ease of reference, we replicate our response below.
> > > > > > > > >
> > > > > > > > > ".......Analysis of applying regularization between $$\hat{X}_{t-1}$ and $X_0$, from two distinct perspectives.
> > > > > > > > >
> > > > > > > > > **a) Trajectory Curvature.** Recent works [C1, C2] show that an efficient and effective ODE-based generative model, e.g., diffusion models, should have a *lower curvature (straight) trajectory*, which could help them stably converge to high-quality solutions. Our regularization term could indeed minimize the trajectory curvature. We further illustrate such properties by corresponding experimental results of the reverse trajectory curve in the additional pdf file. **b) Accumulated Error.** During inference, each step is established on the previous prediction. Thus, accumulated errors inevitably exist in the intermediate reconstruction. Through regularizing $$\hat{X}_{t-1}$ towards the ground truth, we expect that each reconstruction step would consistently move towards GT images, which may also potentially minimize the influence of such accumulated error.
> > > > > > > > >
> > > > > > > > > **2) Analysis of choosing rank-based global-structure aware regularization.**
> > > > > > > > > While the current approaches in image processing often involve pixel-wise treatments, inherent global or latent structures - typically represented by non-local similar patches - can be overlooked by existing models to some extent. Acknowledging and modeling such global structures potentially augments the various reconstruction algorithms' performance [C3]. Additionally, conventional pixel-wise regularization terms, such as L1, L2, and SSIM, do not adequately encapsulate an image's nonlocal structures or patterns. Although regularization within the feature domain potentially aids in modeling a given image's structure, such structure is usually confined within a local region due to the kernel perceptual regions. This type of regularization, moreover, cannot explicitly characterize the properties of the structure and is lacking in theoretical guidance. Another significant concern is that regularization features could result in significant fluctuations in the back-propagated gradients, thereby impeding network training.
> > > > > > > > >
> > > > > > > > > [C1] S. Lee, et al., Minimizing trajectory curvature of ode-based generative models, ICML'23.
> > > > > > > > >
> > > > > > > > > [C2] X. Liu, et al., Flow straight and fast: Learning to generate and transfer data with rectified flow,  ICLR'23.
> > > > > > > > >
> > > > > > > > > [C3] S. Gu, et al., Weighted nuclear norm minimization with application to image denoising, CVPR'14.
> > > > > > > > > ......."
> > > > > > > > >
> > > > > > > > > **Global-structural modeling V.S. deep learning techniques**. The authors agree that rank-based structural modeling was prevalent before the deep learning era. With the emergence of deep learning techniques, those methods become less popular. However, each method has its uniqueness and strength. It's more plausible for us to make use of each potential to build a robust and effective method. How to make use of past mature knowledge/ techniques to direct or boost current work also show very significant importance for the research.
> > > > > > > > >
> > > > > > > > > **Citation** We will ensure further citations are incorporated where indicated. Our deep respect for preceding research is evinced by our extensive list of references, spanning three pages. Furthermore, we consistently strive to find relevant work, regardless of whether it's concurrent or from earlier studies.
> > > > > > > > >
> > > > > > > > > **Finally, we hope our work could give somewhat inspirations to the latent structure in the diffusion process and offer potential avenues for designing regularization terms. We appreciate your further consideration. THANKS.**

---

### Official Review · Reviewer_U5ii · 2023-07-05

**Soundness:** 3 good
**Presentation:** 3 good
**Contribution:** 4 excellent
**Rating:** 7
**Confidence:** 5

**Summary:**

This paper presents an innovative and efficacious approach to regularization within the domain of diffusion models, with particular utility for low-light image reconstruction. The method ingeniously takes into account the latent global structure of images. To achieve this, an image is first divided into patches, and clustering algorithms are then employed to group analogous patches, where the matrix rank is further utilized to minimize the discrepancy between the intermediate reconstruction and the final output. Besides, uncertainty is used to boost performance.

**Strengths:**

1) The novel perspective opens up a promising avenue for enhancing the performance of diffusion models, representing an important contribution to the field. Also, it may have great potential for other diffusion-based image/video processing tasks.
2) The introduction of this form of regularization is a groundbreaking shift for diffusion models, distinguishing this work from existing literature.
3) The experimental validations lend strong support to the efficacy of the proposed regularization term, and the proposed method achieves the current SOTA.
4) The paper is well-written, and the supplementary material includes the code.

**Weaknesses:**

See the detailed questions.

**Questions:**

1) Are other advanced clustering algorithms more effective in enhancing reconstruction performance? This comparison could provide valuable insights into the optimal algorithm selection for this specific task.
2) For the ablation studies in Table 5 (Figure 5 should be Table 5), the authors are suggested to add two more settings, i.e., remove  (b) and (c) and keep (a); remove (c) and keep (a) and (b), to directly validate the adaptive rank regularization, which is the main contributions.
3) In Figure 6, please add the input and GT images. It is better to use different symbols to distinguish the learnable closed-form samples and the input of the forward process.
4) Can other regularization terms (e.g., the L1 loss between X_0 and X_t) improve the original diffusion model?
5) There is potential for the proposed reconstruction algorithm to benefit diffusion models in tasks beyond the current scope of the study. It would be beneficial for the authors to explore this further, extending the validations to other potential applications. At least the authors can discuss this issue to improve the paper.
6) There are some grammar errors, e.g., line 163: a  an

**Limitations:**

Maybe a little additional computation consumption during the training phase.

---

> ### Author Rebuttal · Authors · 2023-08-09
>
> ### **[Responses to Reviewer U5ii]**
>
> ### ***1** Response to Weakness 1 (W1): Clustering Algorithm*
>
> Thanks for your valuable comment. As shown in the following table, our method benefits from advanced clustering algorithms greatly, and the PSNR value is further improved by 0.7 dB, compared with K-means adopted in our manuscript. These advancements further underscore the prospective utility of our regularization terms in enhancing diffusion models.
>
> | Clustering algorithms   | PSNR  | SSIM  | LPIPS |
> |:-------------------------:|:------|-------|-------|
> | K-Means                 | 27.02 | 0.872 | 0.097 |
> | Spectral clustering     | 27.41 | 0.876 | 0.095 |
> | Gaussian Mixture Model  | 27.55 | 0.877 | 0.095 |
> | Hierarchical clustering | 27.70 | 0.880 | 0.092 |
>
> ### ***2** Response to W2: Ablation Settings*
>
> Thanks for the comments. We experimentally verified the importance of our adaptive rank regularization according to the suggested settings. As shown in the following table, our approach demonstrates a significant boost in the baseline, elevating it from 26.02 dB to 27.02 dB, when eliminating (c) uncertainty-guided regularization. This outcome strongly suggests that our major contribution lies in the introduction of adaptive rank regularization.
>
> | (a)      | (b)      | (c)      | PSNR  | SSIM   | LPIPS  |
> |----------|----------|----------|:------|:-------|:-------:|
> | ✕ | ✕ | ✕ | 26.02 | 0.8593 | 0.1226 |
> | ✓ | ✕ | ✕ | 26.63 | 0.8701 | 0.1016 |
> | ✓ | ✓ | ✕ | 27.02 | 0.8722 | 0.0973 |
> | ✓ | ✓ | ✓ | 27.34 | 0.8739 | 0.0969 |
>
> ### ***3** Response to W3: Symbol Issue*
>
> Thanks for the comments. we will certainly add corresponding images to have a better comparison. For symbolic representations, we intend to modify the learnable closed-form sample to enhance its difference from the input.
>
> ### ***4** Response to W4: Other Regularization Terms*
>
> As experimentally verified in the table of the **second response to Reviewer gyFW**, other regularization terms, such as L1 and L2, can also enhance the baseline, e.g.,  the L1 regularization could improve the baseline diffusion model from 26.02 dB to 26.45 dB, and the L2 regularization from 26.02 dB  to 26.53 dB. However, their effectiveness is still much lower than our rank-based regularization improving the baseline diffusion model from 26.02 dB to 27.02 dB, because both L1 and L2 regularization manners fail to fully capture nonlocal structures or patterns within an image and cannot explicitly characterize the properties of the structure. We also refer the reviewer to the response for more detailed analysis.
>
> ### ***5** Response to W5: General Validation*
>
> As shown in the **first response to Reviewer 9ew6**, we also validated the effectiveness of the proposed regularization terms on the image super-resolution task. Upon experiments, it is evident that, by incorporating the proposed regularization term, our approach significantly surpasses the state-of-the-art (SOTA) diffusion SR method by a margin of 0.51 dB. It is widely recognized that SR represents a cornerstone in the domain of image reconstruction. Consequently, this enhancement underscores the profound effectiveness and potential of our methodologies. Besides, in the conclusion section of the final version, we will discuss its potential in other low-level image processing tasks as future work.
>
> ### ***6** Response to W6: Typos*
>
> Thanks for the comments. We will correct those typos in the final version.

---

> > ### Comment · Reviewer_U5ii · 2023-08-15
> > **Response to the rebuttal**
> >
> > I greatly appreciate the authors for their exceptional efforts during the rebuttal phase. I am pleased to note that all of my concerns have been thoroughly addressed. The remarkable achievement of significantly enhancing performance through the utilization of advanced clustering algorithms for grouping image patches in adaptive rank-based regularization is truly impressive. I am also impressed by the authors' clear and well-justified explanations regarding the motivation behind the proposed method, as highlighted in their second response to **Reviewer gyFW**, as well as the superior performance demonstrated compared to state-of-the-art methods.
> >
> > Additionally, after carefully reviewing the comments from my fellow reviewers, I find that the authors' responses effectively address all the questions raised. Drawing from my five years of experience in this field, I hold great confidence in the novelty of the proposed method and believe that its elevated performance level will significantly contribute to the advancement of this research area. Therefore, I believe the paper quality sufficiently meets NeurIPS's standard although I also would like to see the other reviewers’ responses.
> >
> > Furthermore, as previously mentioned, I am convinced that the idea presented in this paper holds potential for other diffusion-based low-level image/video processing tasks, which is somewhat validated by the additional experiments conducted on image super-resolution, as detailed in the authors' response to **Reviewer 9ew6**.
> >
> > In preparing the final version, I highly recommend the authors incorporate the necessary responses as discussed during the rebuttal phase.

---

> > > ### Author Response · Authors · 2023-08-16
> > >
> > > We sincerely appreciate your recognition of our work and effort. We will ensure that the final version will include all the necessary information to make it comprehensive and complete. Thanks.

---

### Official Review · Reviewer_9ew6 · 2023-07-05

**Soundness:** 3 good
**Presentation:** 3 good
**Contribution:** 2 fair
**Rating:** 5
**Confidence:** 5

**Summary:**

Aiming at better low-light image enhancement performance with diffusion models, this paper proposes a global structure-aware regularization utilizing the intrinsic non-local structural constituents of image data. An uncertainty map is incorporated into the diffusion model to ease the strict constraints on indeterminate regions. Experimental results demonstrate the effectiveness of the proposed components.

**Strengths:**

1.	The paper incorporates global structure regularization via matrix rank modeling into the diffusion models, showing significant performance improvement compared with state-of-the-art methods.
2.	The paper is well-written, and the presentation is pretty good.


**Weaknesses:**

1.	The contribution is not convincing enough. The generality of the proposed global structure-aware regularization based diffusion models is not verified, and the main idea of the adopted uncertainty is not original in this paper.
2.	Additionally, the main concern about this paper is that there are many unclear details. For example, what train data is used in this paper? What is the baseline setting in Table.5? Is the inference time of all the methods also performed on RTX 3080 GPUs?
3.	More perceptual metrics (e.g., LPIPS, MUSIQ) could be reported for better evaluation.


**Questions:**

Please see the strengths and weakness section.

**Limitations:**

Several challenging cases are provided. The limitations of the proposed approach sholud be discussed in more detail.

---

> ### Author Rebuttal · Authors · 2023-08-09
>
> ### **[Responses to  Reviewer 9ew6]**
>
> ### ***1** Response to W1: Paper Contributions*
>
> Thanks for the comments. We conducted additional experiments in terms of image super-resolution ($16\times$ SR on CelebA-HQ dataset) to validate the generality of our method on different tasks.
> The quantitative results in the following table show that our global structure-aware regularization can improve the reconstruction performance, demonstrating its generality to some extent. The results of the SOTA baseline [C1] are also provided for comparison.
>
> | Methods                                        | PSNR  | SSIM  |
> |------------------------------------------------|-------|-------|
> | IDM [C1]                                       | 24.01 | 0.710 |
> | Ours w/o global structure-aware regularization | 23.94 | 0.713 |
> | Ours                                           | 24.52 | 0.728 |
>
> Besides, it is also worth mentioning that as discussed in the **first response to Reviewer U5ii**, the performance of our method is further improved significantly by applying advanced clustering methods for grouping non-local patches, i.e., the overall PSNR value increases about 0.7 dB, compared to the other clustering algorithms. We also refer the reviewer to the **second response to Reviewer gyFW** for the detailed analysis of our method and additional experimental results of various regularization terms.
>
> In all, we posit that our contributions will pave the way for fresh perspectives on diffusion models for low-level image processing tasks. And our method improves the SOTA performance of low-light image enhancement to a new higher level, which will contribute to the community.
>
> [C1] S. Gao, et al., Implicit diffusion models for continuous super-resolution, CVPR'23.
>
>
> ### ***2** Response to W2: Setting Details*
>
> Sorry for the confusion caused. We will complement the details.
>
> 1) We kindly remind the reviewer to refer to Section 4.1 of the manuscript for the details of the used datasets. For example, we utilized 485 pairs of low/normal-light images for training and 15 pairs for testing on the LOLv1 dataset.
> 2) The baseline was constructed by removing the non-local patch-based matrix rank modeling and uncertainty-guided regularization. The baseline corresponds to the basic diffusion model.
> 3) The inference time per image of all the recent SOTA methods in Table S1 of Supplementary Material was conducted using an RTX3080 GPU on the same server.
>
> ### ***3** Response to W3: Comparison of More Perceptual Metrics*
>
> In the manuscript, we have indeed provided the LPIPS comparisons in Table 1. For better evaluation, we further adopted MUSIQ (trained on KonIQ-10k) for assessing the enhancement quality. The following table shows quantitative comparisons of the recent SOTA methods in terms of MUSIQ, which consistently demonstrate the superiority of our method.
>
> | Datasets        | SNR-Aware         | LLFormer | Ours      |
> |--------------|-----------------|---------|--------|
> | LOLv1           | 61.00             | 58.26    | **71.74** |
> | LOLv2-real      | 57.76             | 51.13    | **69.34** |
> | LOLv2-synthetic | 63.18             | 62.84    | **64.51** |

---

> > ### Author Response · Authors · 2023-08-16
> > **Looking forward to hearing from you**
> >
> > Dear **Reviewer 9ew6**
> >
> > Thank you for taking the time to review our submission and providing us with constructive comments and a favorable recommendation. We would like to know if our responses adequately addressed your earlier concerns. Additionally, if you have any further concerns or suggestions, we would be more than happy to address and discuss them to enhance the quality of the paper. We eagerly await your response and look forward to hearing from you.
> >
> > Best regards,
> >
> > The authors

---

> > ### Comment · Reviewer_9ew6 · 2023-08-17
> >
> > Thanks for the rebuttal. For W2-1, my concern is whether the proposed method is trained only on LOL-v1 or separately on LOL-v1 and LOL-v2 due to the ambiguity stated in Section4.1. If the latter is the case, then for fair comparison, the methods in Table1 that only provide pretrained weights on LOL-v1 should be pointed out separately. Also, will you release the source code to ensure reproducibility?

---

> > > ### Author Response · Authors · 2023-08-17
> > >
> > > We appreciate the feedback received. In accordance with the settings outlined in **the very recent** works, i.e., SNR-Aware (CVPR'22), LLFlow (AAAI'22), and LLFormer (AAAI'23), we also trained our models on LOLv1 and LOLv2 datasets separately. Regarding the remaining methods, we derived the most promising results from relevant papers (e.g., LLFlow, LLFormer, and SNR-Aware), ensuring consistent settings for fair comparisons. We will make this clear in the final version. Besides, it is worth noting that **we have conscientiously included the source code within the supplementary material**. Additionally, we will make our code and pre-trained models publicly available, with the aim of facilitating others in reproducing our results.

---

### Official Review · Reviewer_gyFW · 2023-07-05

**Soundness:** 2 fair
**Presentation:** 2 fair
**Contribution:** 2 fair
**Rating:** 4
**Confidence:** 5

**Summary:**

This work proposes a diffusion-based low-light image enhancement framework that exhibits good performance in different benchmarks. A rank-informed regularization term during training and uncertainty-weighted loss is proposed.

**Strengths:**

- This work proposes a diffusion-based low-light image enhancement framework that seems to exhibit good performance in different benchmarks.




**Weaknesses:**

- The motivation of the "Non-local patch-based matrix rank modeling" seems not clear.

According to my understanding, the proposed "Non-local patch-based matrix rank modeling" is only used during training. How does it make the model have the capacity to be aware of the global structure during inference?

Besides, during the training using paired training data, what's the necessity of using the proposed method to measure the structure consistency between Xt−1 and X0? The difference be directly calculated using metrics like L1, L2, SSIM, or feature-level similarities.

- How does the proposed method increase the training overhead?

- Some formulation is not self-contained. For example, where is the definition of "\alpha_1^2" in Eq. (4)?  How to set the value of k_t?

- It's better to illustrate which dataset/subset of the dataset is used to train the proposed method and other competitors. Since it greatly affects the performance of the model. And including a comparison of inference cost is better.


**Questions:**

- Please refer to the weakness section.

- While the weighting strategy of the training loss is reasonable for idea cases. But for real-captured high-resolution images where small misalignments may exist, will it lead to some side effects?
- Could you explain why the constraint need only be added to the singular value, while no need for the singular vector?

---

> ### Author Rebuttal · Authors · 2023-08-09
>
> ### **[Responses to Reviewer gyFW]**
>
> ### ***1**  Response to Weakness 1 (W1): Relation between Global Structure Modeling and Non-local Patch Rank Modeling*
>
> It is imperative to highlight that by "global structure-aware," we are referring to the network's ability to account for various patterns, such as non-local patches, rather than solely focusing on individual pixels. By regularizing the matrix rank formed from a cluster of these non-local patches, our model can concurrently consider structures from various positions, resulting in a strong capability of global structure modeling.
>
> Second, the characteristics and properties of a neural network are primarily acquired during the training phase. In our approach, we incorporate specific regularization to encourage the network weights to reconstruct the global structure. During inference, the network can still effectively exhibit the learned function, by the established weight pattern in the training phase.
>
> ### ***2**   Response to W2: Theoretical Soundness*
>
> We answer this question from two levels: why apply regularization between $X_{t-1}$ and $X_0$, and then why choose the rank-based global structure-aware regularization.
>
> **1) Analysis of applying regularization between $X_{t-1}$ and $X_0$, from two distinct perspectives.**
> a) **Trajectory Curvature**. Recent works [C1, C2] show that an efficient and effective ODE-based generative model, e.g., diffusion models, should have a **lower curvature (straight) trajectory**, which could help them stably converge to high-quality solutions. Our regularization term could indeed minimize the trajectory curvature. We further illustrate such properties by corresponding experimental results of the reverse trajectory curve in the additional pdf file.
> b) **Accumulated Error**. During inference, each step is established on the previous prediction. Thus, accumulated errors inevitably exist. Through regularizing $X_{t-1}$ towards the ground-truth, we expect that each reconstruction step would consistently move towards GT images, which may also potentially minimize the influence of such accumulated error.
>
> **2) Analysis of choosing rank-based global-structure aware regularization**
> While the current approaches often involve pixel-wise treatments, inherent global structures can be overlooked to some extent. Modeling such global structures potentially augments the performance [C3]. Additionally, pixel-wise regularization terms, such as L1, L2, and SSIM, do not adequately encapsulate nonlocal structures. However, regularization within the feature domain is usually confined within a local region due to the kernel perceptual regions.  Moreover, regularizing features can result in fluctuations in the back-propagated gradients, thereby impeding network training.
>
> Finally, the following table also verifies the above analyses, i.e., the necessity of introducing regularization between $X_{t-1}$ and $X_0$, and the advantage of our rank-based model.
>
> |Method| PSNR| SSIM| LPIPS|
> |--|--|--|--|
> | Baseline| 26.02| 0.859| 0.123 |
> | L1 Reg. | 26.45| 0.868| 0.101 |
> | L2 Reg. | 26.53| 0.869| 0.102 |
> | SSIM Reg.| 26.23| 0.870| 0.101|
> | Perceptual Reg.| 26.42| 0.863| 0.099|
> | Rank Reg. (Ours) | **27.02** | **0.872** | **0.097** |
>
> [C1] S. Lee, et al., Minimizing trajectory curvature of ode-based generative models, ICML'23.
>
> [C2] X. Liu, et al., Flow straight and fast: Learning to generate and transfer data with rectified flow,  ICLR'23.
>
> [C3] S. Gu, et al., Weighted nuclear norm minimization with application to image denoising, CVPR'14.
>
> ### ***3**  Response to W3: Training Overhead*
>
> Our method significantly improves the quality of enhanced images by exploring global structures. Still, it simultaneously introduces little additional computational consumption, only about 0.03 extra seconds (from 0.105 to 0.135 S) to train for one iteration on the LOLv1 dataset. Besides, no additional computational cost is introduced during inference.
>
>
> ### ***4**   Response to W4: Notions*
>
> Sorry for the confusion caused. $\alpha_t$ in Eq. (4) is the same as the $\alpha_t$ in Eq. (2) and **Algorithm 1**, which is defined in Line 88. Moreover, $k_t$ means $\alpha_t^2$. We will clarify them in the final version.
>
> ### ***5**   Response to W5: Training Dataset*
>
> In alignment with existing methods under comparison, we conducted the network training utilizing the respective training datasets. For instance, during the LOLv1 testing experiment, we base the network's training on the LOLv1 training set. Additionally, for a comprehensive understanding of the inference cost, **we kindly refer the reviewer to the first section of the supplementary.**
>
> ### ***6**   Response to Question 1 (Q1): Treating Misalignments*
>
> With misalignments, a decline in reconstruction performance across all methods might be observed. However, generative models, such as diffusion models, just rely on conditioning from input images rather than directly translating inputs to the reconstruction, as seen in regression-based methods. Therefore, their requirements for alignment might be less stringent. Furthermore, the uncertainty module in our method may potentially address misalignments by moderating the regularization on misaligned pixels.
>
> ### ***7**   Response to Q2: Why not Regularize Singular Vectors*
>
> We regularize the global structure consistency between two images. With such a regularization term, it is expected that the similarity of different non-local patches in $X_{t-1}$ and $X_{0}$ should be identical, or a same component proportions of different images. Thus, by approaching singular values, we could minimize the gaps between $X_{t-1}$ and $X_{0}$. Note that simultaneously regularizing the singular values and vectors to be the same is equal to regularizing all patches to be pixel-wise similar. Moreover, as listed in **the second response to Reviewer gyFW**, the L1 regularization produces inferior performance. Thus, we do not further regularize singular vectors.

---

> > ### Author Response · Authors · 2023-08-16
> > **Looking forward to hearing back from you**
> >
> > Dear **Reviewer gyFW**
> >
> > Thank you for taking the time to review our submission and providing us with constructive comments. We would like to inquire if our responses have adequately addressed the concerns you raised earlier. Additionally, if you have any further concerns or suggestions, we would be more than happy to address and discuss them in order to enhance the quality of the paper. We eagerly await your response and look forward to hearing from you.
> >
> > Best regards,
> >
> > The authors

---

> > ### Comment · Reviewer_gyFW · 2023-08-19
> >
> > Thanks for your reply.
> >
> > How did you test the LOLv2-real dataset since according to the attached code the model is not adapted for the image with the resolution 400*600?
> >
> > Besides, the name of the released checkpoint is "LOLv2-syn.pth". Does it mean you train two separate models for LOLv2-real and LOLv2-syn?

---

> > > ### Author Response · Authors · 2023-08-20
> > > **Official Comment by Authors**
> > >
> > > We appreciate the feedback received. For testing on LOLv2-real, we resized the input image (400×600) by padding it to the size of 416×608 and cropped the output back to its original dimensions. Besides, we indeed train two separate models for the LOLv2-real and LOLv2-syn datasets, consistently following the settings outlined in the very recent work, i.e., SNR-Aware (CVPR'22), which conducted extensive experiments on these datasets. Lastly, we will update and make our code and pre-trained models publicly accessible to facilitate the reproduction of our results by others. We sincerely thank you once again for the efforts to assist in improving the quality of our work.

---

### Official Review · Reviewer_ts6H · 2023-07-11

**Soundness:** 3 good
**Presentation:** 3 good
**Contribution:** 2 fair
**Rating:** 5
**Confidence:** 5

**Summary:**

This paper presents a diffusion-based framework to enhance low-light images. They propose a global structure regularization, which leverages the intrinsic non-local structural constituents of image data, besides, they introduce an uncertainty-guided regularization technique, which relaxes constraints on the most extreme portions of the image. The result outperforms some SOTA methods in some low-light image enhancement datasets.

**Strengths:**

The paper proposes a global structure-aware regularization scheme, which capitalizes on image intrinsic non-local structures and gradually adjusts the regularization strength according to the sampling steps.

The method achieves good results in the LOL dataset.

**Weaknesses:**

My main concern is about the novelty of the method, the uncertainty-guided regularization seems from[4]

 Although the author has done comparison experiments in some datasets. I think the number of these images is quite limited, I hope the paper can provide a comparison experiment in the Adobe-FiveK dataset[1], following the setting of[2]

The method needs to resample 500 steps. it will cost a long time to infer an image.

The paper should cite and compare with[3].


[1] Learning Photographic Global Tonal Adjustment with a Database of Input / Output Image Pairs
[2] Underexposed Photo Enhancement Using Deep Illumination Estimation
[3]Pyramid Diffusion Models For Low-light Image Enhancement
[4]. Uncertainty-driven loss for single image super-resolution.

**Questions:**

Why does the method choose DDPM? How about DDIM?

**Limitations:**

See in weakness.

---

> ### Author Rebuttal · Authors · 2023-08-09
>
> ### **[Responses to Reviewer ts6H]**
>
> ### ***1** Response to Weakness 1 (W1): Novelty of the Method*
>
> We would like to underscore that the **MAJOR** innovation presented in our paper is the integration of global structure regularization into diffusion models.  We also refer the reviewer to the **second response to Reviewer gyFW** for detailed analysis. Furthermore, by leveraging sophisticated clustering algorithms for grouping non-local patches, the overall PSNR of our method is further improved to 0.7 dB, convincingly attesting to the importance and efficacy of regularizing global structures (see **the first response to Reviewer U5ii**.  We posit that our contributions will pave the way for fresh perspectives on diffusion models for low-level image processing tasks, as discussed by **Reviewer U5ii**.
>
> In the manuscript, we have indeed acknowledged that our work is inspired by [4]. As the first time, we demonstrate that uncertainty-aware regularization is a simple yet effective way to boost the performance of the diffusion process under low-light image enhancement. But we also agree that this contribution is somewhat minor. We believe the major contribution meets the NeurIPS's standard based on its novelty and impressive performance.
> In the final version, we will re-summarize the contribution part to emphasize the key contribution and list the uncertainty as a minor contribution.
>
> ### ***2** Response to W2: Evaluation Dataset*
>
> Following your valuable suggestion, we conducted the experiments on the Adobe-FiveK dataset in the same setting as [2]. The quantitative results shown in the following table still verify the advantages of our method. We will add further discussion and review of [1,2] in the final version.
>
> | Methods      | PSNR  | SSIM  |
> |--------------|-------|-------|
> | DeepUPE [2]  | 23.04 | 0.893 |
> | Ours         | 23.77 | 0.912 |
>
> ### ***3** Response to W3: The Number of Inference Steps*
>
> Our diffusion model takes only **10 steps for inference**, as described in supplementary material. 500 steps are only utilized for training. It is essential to recognize that the use of multiple sampling steps is not an attribute exclusive to our method, but rather a common issue across all diffusion methods. Moreover, some methodologies have been developed with the aim of reducing the steps required within the diffusion model. In addition, it is pivotal to highlight that the inference method employed in our methods operates in alignment with a standard diffusion model. Consequently, this allows for the integration and utilization of advanced fast-sampling techniques, as detailed in references [C1].
>
> [C1] C. Lu, et al., Dpm-solver: A fast ode solver for diffusion probabilistic model sampling in around 10 steps, NeurIPS'22
>
> ### ***4** Response to W4: Related Work*
>
> It is worth noting that the mentioned work [3] was accepted in IJCAI'23 and first available in arXiv on 2023 May 17, which definitely coincided with the paper submission deadline for NeurIPS'23. Thus, it should be considered as a concurrent work with ours. But we made comparisons with it as its test code is available. Under the same test settings as ours, we obtained the results on LOLv1 by running the publicly released pre-trained model that was trained with the same training dataset as ours. As shown in the following table, the advantage of our method is still verified.
>
> | Methods    | Architecture    | Loss term                                                 | PSNR   | SSIM  | LPIPS |
> |------------|---------------|------------------------------------------|------|-----|----|
> | PyDiff [3] | Pyramid         | Vanilla with Multi-scale L1                               | 27.090 | 0.879 | 0.100 |
> | Ours       | Vanilla (U-Net) | Rank-based modeling with basic KMeans clustering          | 27.336 | 0.874 | 0.097 |
> | Ours++     | Vanilla (U-Net) | Rank-based modeling with advanced Hierarchical clustering | **27.697** | **0.880** | **0.092** |
>
> Additionally, we also want to note that our approach enhances the diffusion process through a plug-and-play regularization term rather than the modification of the network structure adopted in [3]. As a result, there exists potential for integrating our regularization term into [3] to achieve superior reconstruction. We anticipate exploring this integration once the training code from [3] is released.
>
> ### ***5** Response to Question 1 (Q1): DDPM or DDIM*
>
> DDPM has shown its great capability in producing high-quality images, to explore the effectiveness of our global structure-aware regularization within diffusion models,  we thus integrate our regularization terms into this foundational diffusion model,  exhibiting superior performance relative to state-of-the-art (SOTA) methods. This clearly highlights the potential of our approach. With regard to DDIM, it aims to accelerate the sampling process by generalizing DDPM using non-Markovian diffusion processes. In essence, DDIM can be considered as a generalized version of DDPM, and they have the same training procedure. Consequently, we are confident that our regularization terms could be seamlessly integrated into DDIM, offering significant potential for achieving exceptional performance.

---

> > ### Author Response · Authors · 2023-08-16
> > **Looking forward to hearing from you**
> >
> > Dear **Reviewer ts6H**
> >
> > Thank you for taking the time to review our submission and providing us with constructive comments and a favorable recommendation. We would like to know if our responses adequately addressed your earlier concerns. Additionally, if you have any further concerns or suggestions, we would be more than happy to address and discuss them to enhance the quality of the paper. We eagerly await your response and look forward to hearing from you.
> >
> > Best regards,
> >
> > The authors

---

> > ### Comment · Reviewer_ts6H · 2023-08-16
> >
> > Thanks for your rebuttal. I think you should add a comparison with MAXIM: Multi-Axis MLP for Image Processing(CVPR' 22), which can achieve 26+ PSNR on the AdobeFiveK. It seems to have a 2+ db gap.

---

> > > ### Author Response · Authors · 2023-08-16
> > >
> > > We appreciate the feedback received. In light of the previous comments, we conducted a comparative analysis in alignment with the original configuration proposed by [2]. Our evaluation encompassed **all 500 test images at their native resolutions**. It's noteworthy that MAXIM compared **only 400 test images and subjected them to cropping and resizing to the resolution of $512 \times 512$**. Consequently, **drawing a fair comparison with their published results proves difficult**. Given our time constraints, our present model on the FiveK dataset has been adjusted in a preliminary manner. With a more meticulous calibration, there is potential for enhancing our results further.
> > >
> > > It is also worth mentioning that we are devising global structure-aware loss terms to explore the potential of the diffusion model. Thus, advanced backbones, e.g., the mentioned MAXIM, could be integrated into our framework to replace the vanilla U-Net for further performance improvement.

---

### Author Rebuttal · Authors · 2023-08-09

### General Purpose

We thank all reviewers for your time, constructive comments, and recognition of our work. We believe all concerns have been clearly and directly addressed. Here we also want to summarize a few key clarifications concerning the contributions of our work.

Our **MAJOR** contribution lies in introducing an adaptive rank-based regularization term to promote the baseline diffusion model to capture the global structure. Specifically, we aim to minimize the discrepancy between $X_{t-1}$ and $X_{0}$ to establish the reverse trajectory with reduced curvature, which can potentially benefit high-quality samples generation [C1, C2]. Moreover, during this process, we incorporate rank-based modeling focused on non-local similar patches, thereby exploiting the global structure modeling capabilities in diffusion models. We have experimentally and precisely validated the necessity of regularization between $X_{t-1}$ and $X_{0}$, as well as the effectiveness of rank-based modeling. See the figures in the submitted pdf file and the second response to **Reviewer gyFW** for the detailed analyses. Besides, as experimentally verified in the **first response to Reviewer U5ii**, by simply leveraging advanced clustering algorithms for grouping image patches, the PSNR value of our approach further increases about 0.7 dB, emphasizing the potential utility of our regularization terms in enhancing diffusion models.

We posit that our contributions will pave the way for fresh perspectives on diffusion models for low-level image processing tasks. And our method improves the SOTA performance of low-light image enhancement to a new higher level, providing a promising benchmark to this community.

Last but not least, **we will make the reviews and author discussion public regardless of the final decision**.  Besides, we will include the newly added experiments and analysis in the final manuscript/supplementary material.

---

> ### Comment · Area_Chair_pvFy · 2023-08-13
> **Thanks for the rebuttal.**
>
> Thanks for the rebuttal. The AC and reviewers will take this into account for the final rating.

---

### Decision · Program_Chairs · 2023-09-21

**Decision:**

Accept (poster)

**Comment:**

This paper presents a novel regularization approach within the domain of diffusion models for low-light image enhancement. The method utilizes the rank of clustered non-local image patches as a measure of global structure. Experimental results show good performance. The paper received five reviews. Most of the reviews are positive, except that one reviewer has concerns on its applications to other low-level image enhancement tasks (other than low-light images) and the issues with the paper writing/citation.

The author did a thorough rebuttal and carefully addressed most of the reviewers' question. The main contribution, i.e., introducing the rank-based constraint to enforce global structure for diffusion-based image restoration seems novel and effective. In the rebuttal, the authors also tried advanced clustering algorithms as well as other image restoration tasks such as super-resolution, and the results are good. While the AC agreed this rank-based regularization can be used for general image restoration tasks, the paper itself is self contained as a complete work, considering the novelty in the idea, the SOTA experimental results in low-light image enhancement, and the added results and analysis in the rebuttal.

Taking into account all the reviews, the rebuttal, and the discussions, the AC decided to accept the paper.